# A Novel Pulmonary Nodule Detection Model Based on Multi-Step Cascaded Networks

**DOI:** 10.3390/s20154301

**Published:** 2020-08-01

**Authors:** Jianning Chi, Shuang Zhang, Xiaosheng Yu, Chengdong Wu, Yang Jiang

**Affiliations:** Faculty of Robot Science and Engineering, Northeastern University, No. 195, Chuangxin Road, Shenyang 110169, China; zs1901968@163.com (S.Z.); yuxiaosheng@mail.neu.edu.cn (X.Y.); wuchengdong@mail.neu.edu.cn (C.W.)

**Keywords:** pulmonary nodule detection, deep neural convolutional network, inception structure, dense connection, dilated convolution, multi-resolution convolution

## Abstract

Pulmonary nodule detection in chest computed tomography (CT) is of great significance for the early diagnosis of lung cancer. Therefore, it has attracted more and more researchers to propose various computer-assisted pulmonary nodule detection methods. However, these methods still could not provide convincing results because the nodules are easily confused with calcifications, vessels, or other benign lumps. In this paper, we propose a novel deep convolutional neural network (DCNN) framework for detecting pulmonary nodules in the chest CT image. The framework consists of three cascaded networks: First, a U-net network integrating inception structure and dense skip connection is proposed to segment the region of lung parenchyma from the chest CT image. The inception structure is used to replace the first convolution layer for better feature extraction with respect to multiple receptive fields, while the dense skip connection could reuse these features and transfer them through the network. Secondly, a modified U-net network where all the convolution layers are replaced by dilated convolution is proposed to detect the “suspicious nodules” in the image. The dilated convolution can increase the receptive fields to improve the ability of the network in learning global information of the image. Thirdly, a modified U-net adapting multi-scale pooling and multi-resolution convolution connection is proposed to find the true pulmonary nodule in the image with multiple candidate regions. During the detection, the result of the former step is used as the input of the latter step to follow the “coarse-to-fine” detection process. Moreover, the focal loss, perceptual loss and dice loss were used together to replace the cross-entropy loss to solve the problem of imbalance distribution of positive and negative samples. We apply our method on two public datasets to evaluate its ability in pulmonary nodule detection. Experimental results illustrate that the proposed method outperform the state-of-the-art methods with respect to accuracy, sensitivity and specificity.

## 1. Introduction

Lung cancer is one of the most lethal diseases with only about 16% 5-year survival rate [1,2]. With the development of modern medical techniques, researchers have proved that the survival rate could achieve 54% on average if the lung cancers could be diagnosed in the early stage [3]. Therefore, early detection of pulmonary nodule plays a critical role in the early diagnosis of lung cancers [4] and computer-assisted diagnosis system (CADs) [5]. Recently, pulmonary nodule detection has been typically performed on the chest CT scans, and many automated detection methods have been proposed by processing and analyzing the chest CT images [6]. These methods can be generally categorized into two types: (1) detection based on hand-crafted features, and (2) detection based on deep learning.

Most traditional pulmonary nodule detection methods extracted hand-crafted features in every region of interest and trained classifiers to judge whether the region contained pulmonary nodules [7]. In [8], Nithila et al. proposed to integrate the intensity cluster, rolling ball and active contour model (ACM) to detect nodule candidates and send the statistical and texture features of these candidates to a back propagation neural network to find the true nodules. Wu et al. proposed to first segment the nodule candidates from the image through thresholding, region growing and morphological operations, then classify these candidates by a support vector machine (SVM) into true or false positive ones. A generic algorithm template matching (GATM) method was proposed by Lee et al. [9] to detect nodules inside the parenchyma area, where shape and gradient features were applied as the classification rules. Farag et al. [10] improved this method by using Gaussian templates whose parameters were estimated based on the given data, achieving higher detection accuracy. These methods could provide good performance when the features and the classifier fit well on the certain CT images. However, the hand-crafted features relied heavily on the prior empirical knowledge from radiologists and were difficult to be optimized with classifiers for images from different sources. Therefore, these hand-crafted features-based methods suffered from limited accuracy and applicability.

In recent decades, deep learning methods have been widely used in various medical image analysis problems, such as de-noising, segmentation and nodule detection. Specifically, for pulmonary nodule detection, Ding et al. [11] took the advantage of three consecutive CT slices as input for the network and applied a faster region-convolutional neural network to locate the nodules in the image. In [12], Dou et al. implemented a three-dimensional fully convolutional neural network (3D-FCN) to detect nodule candidates and integrated two residual blocks for accurate classification of true nodules. Zuo et al. [13] transferred a convolution neural network (CNN) that was applied to edge detection and improved it into a multi-resolution model for image classification task. In [14], Zhao et al., first segmented the lung parenchyma by thresholding and morphological operations, then patched 3D U-Net with adversarial training to localize nodule candidates, and proposed a contextual CNN to classify the candidate nodules. Zhang et al. [15] proposed a 3D progressive resolution-based densely dilated FCN, known as the progressive resolution network (PRN) to detect nodule candidates inside the lung area. Then they proposed a densely dilated 3D CNN with hierarchical saliency, known as the hierarchical saliency network (HSN), to distinguish the true nodules from other candidates and estimate the diameters of nodules simultaneously. Generally, the deep learning-based methods performed better than traditional methods because they could extract features closer to human perception and they unified feature extraction and pattern classification in one framework. However, they had two major shortcomings: (1) there were so many small tissues all over the chest CT image that it was difficult for the traditional DCNN to process every nodule candidate; (2) the convolution kernels in traditional DCNN were with the same size and small receptive fields, resulting in difficulties in distinguishing the subtle differences between true nodules and non-nodule tissues.

To solve the problems of the traditional DCNNs discussed above, in this paper, we propose a novel framework cascading lung parenchyma regions segmentation, nodule candidate detection and true nodule determination to detect pulmonary nodules in the chest CT image. Figure 1 illustrates the flowchart of the proposed model, which consists of the following steps: (1) segment the lung parenchyma regions by a U-Net-like network where inception structure is applied as the first convolution layer and dense connections are used as the skip connection, (2) detect the suspicious nodule regions by a U-Net-like network where the conventional convolutions are replaced by dilated convolutions to enlarge the receptive fields, (3) determine true nodules from nodule candidates by a U-Net-like network adapting multi-scale pooling and multi-resolution convolution connection, (4) unify the three sub-networks as an end-to-end network and train it with modified dice loss, pixel-wise loss and perceptual loss together. We apply our proposed method to the public LUNA16 dataset [16] and the ALIBABA TianChi dataset and achieve superior performance over the state-of-the-art methods with respect to the benchmarks. The advantages and disadvantages of all the above comparison methods, including our proposed method are summarized in Table 1, and we will discuss these methods in detail in our work.

The main contributions of this work are listed as follows:we propose a uniform framework with three hierarchical U-Net-like [17] networks following the “coarse-to-fine” manner: (a) parenchyma region detection, (b) nodule candidate detection, and (c) true nodule determination;we apply the inception structure and dense connection in the U-Net-like network for better segmentation of lung parenchyma regions;we leverage dilated convolutions instead of conventional ones in the U-Net-like network so that those small-size tissues including nodules can be detected without omission;we adapt multi-scale pooling strategy and multi-resolution convolution block in the U-Net-like network to differentiate the true nodules with multiple sizes from the candidates in the complicated environment;we modify the dice loss by considering the imbalance distribution of positive and negative samples. The proposed dice loss, together with the pixel-wise loss and the perceptual loss, are used to train the proposed cascaded network.

The rest of this paper is organized as follows: in Section 2 we describe the three sub-networks for lung parenchyma regions segmentation, nodule candidate detection and true nodule determination in detail, respectively. The joint loss and the training strategy are discussed in Section 3. The experimental results are presented and discussed in Section 4 and finally we conclude our work in Section 5.

## 2. Pulmonary Nodule Detection Framework

### 2.1. Inception-Dense U-Net for Lung Parenchyma Segmentation

Given to the fact that pulmonary nodules only occur in the lung parenchyma, it is necessary to precisely segment the parenchyma region from the CT image to avoid the interference from those outside-lung organs and tissues such as sternums in the nodule candidates’ detection. As shown in Figure 2, we propose a modified U-Net network to find the lung parenchyma mask of the input CT image. An inception [18] module is used to replace the first convolution layer of the original U-Net to extract features using different receptive fields. Moreover, dense connections between all convolution layers and de-convolution layers are used instead of the simple skip connections between each convolution layer and its corresponding de-convolution layer. We will introduce the inception module and dense connection in detail in the following parts.

#### 2.1.1. Inception Structure

Figure 3 shows the inception block we used in our proposed segmentation network. The input layer is filtered by convolution kernels with different sizes in parallel, and the results from these channels are concatenated together. Therefore, more comprehensive features of the outside-lung organs, lung parenchyma and the boundary of pulmonary lobe could be extracted from the image, leading to better segmentation of lung parenchyma. Before connected to next layer, a 1×1 convolution layer is applied to reduce the dimensionality of the feature maps to preserve only the necessary information using as few parameters as possible. The Inception module is only applied as the first convolution layer for a good tradeoff between multi-scale features separation and the computational cost.

#### 2.1.2. Dense Connection

During the forward transmission of the U-Net, image features are very likely to be lost, and the incomplete features will result in incorrect segmentation. Enlightened by the Dense-Net proposed in [19], we propose to connect every convolution layer to all the other convolution layers in the network. Therefore, the feature maps from all the previous layers are used as input the current layer and the feature map from the current layer will be used as the input of all the subsequent layers. It could avoid the problem of feature vanishing and improve the segmentation performance as a result.

As shown in Figure 4, the 4 convolution layers are connected following the dense manner. The relation between these layers can be mathematically represented as follows:(1)$$X_{n}=H_{n}\left(X_{0↓8},X_{1↓4},\cdots ,X_{(n-1)↓2},X_{n}\right),$$
where $X_{n}$ is the feature map from the *n*-th layer, $H_{n}\left(\cdot \right)$ denotes the convolution operation of the *n*-th layer, ↓m represents the *m*-times down-sampling operation of the certain layer. The input of the $X_{n}$ layer are the concatenated features of those from previous n−1 layers after down-sampling. The dense connection between the de-convolution layers follows the opposite way of the convolution ones.

Figure 5 shows the input CT image together with the lung parenchyma mask detected by the first proposed U-Net-like network. It can be considered that the mask fits the ground truth well so those outside-lung tissues, even with similar sizes and shapes as pulmonary nodules, are filtered out from the input of the next stage.

### 2.2. Dilated-Convolution U-Net for Nodule Candidate Detection

Taking the image with outside-lung regions filtered out as input, we propose another U-Net-like network to detect nodule candidates. The structure of the detection network is shown as Figure 6, where every convolution layer is replaced by the convolution block consisting of inception module and dilated convolution, while every de-convolution layer is followed by dilated convolution to form the de-convolution block.

The traditional U-Net makes use of pooling layer with regular convolution and the pooling stride larger than 1 to increase the receptive field for global features learning. However, this operation decreases the size of the feature map, resulting in degradation of resolution, loss of important information, and low detection accuracy after up-sampling to the same size of input image. The dilated convolution [20] is applied to solve this problem by introducing a parameter called “dilation rate” to represent the distance between every two non-zero weighted parameters in the convolution kernel. The sizes of the dilated-convolution kernel and the feature map after convolution can be calculated as follows:(2)$$\begin{array}{cc} n & =k+(k-1)\times (d-1)\\ o & =\frac{i-2p-n}{s}+1 \end{array},$$
where *k* represents the kernel size, *p* denotes the number of 0 during the convolution, *s* is the convolution stride, *d* is the dilation rate, *i*, *n* and *o* are the size of input feature map, new kernel after dilation and output feature map, respectively. It can be considered that the convolution operation could contain larger range of information without pooling operation.

Figure 7 shows the general structure of the convolution block used in Figure 6. The inception module is used to extract the features in multiple perceptual scales as discussed in Section 2.1.1. After that, the dilated convolution is used instead of pooling operation to maintain the receptive field so the output feature map can preserve differences between nodule candidates and other tissues as much as possible.

Contrarily, the structure of the de-convolution block is shown in Figure 8. After the concatenation of de-convolution results and the corresponding convolution results, two dilated convolutions are used continuously. Similar to those used in convolution steps, dilated convolutions in the de-convolution block can preserve the perceptual information more than the traditional up-sampling. Figure 9 shows the visualization of the output of the second proposed U-Net-like network, where nodule candidates can be detected comprehensively without missing those true nodules.

### 2.3. Multi-Resolution Feature Concatenation and Multi-Scale Pooling CNN for Pulmonary Nodule Determination

The lung parenchyma and the nodule candidate regions are detected by the two U-Net-like networks discussed in Section 2.1 and Section 2.2, respectively. In this part, we propose the third network to differentiate pulmonary nodules from non-nodule candidates, which can be considered to be an expansion of the second network. Figure 10 shows the structure of the proposed network for classification, where the multi-resolution feature concatenation module is used to replace the conventional convolution layers and the multi-scale pooling is applied to replace the max-pooling. Therefore, features of the small nodule candidates can be extracted more clearly for better classification of true pulmonary nodules.

#### 2.3.1. Multi-Resolution Convolution Block

Inspired by the HRNet proposed in [21], as shown in Figure 11, we propose a multi-resolution feature concatenation module to extract more detailed features of those small-size nodule candidates. The input layer is processed by convolution kernel with 3 different strides in parallel:stride equals to 1;stride equals to 1 in horizontal direction and 2 in vertical direction;stride equals to 2 in horizontal direction and 1 in vertical direction.

It then generates the feature maps in 3 different resolutions:feature map with the same high resolution as the input layer;feature map with the 1/2 resolution in horizontal direction while the same resolution in vertical direction as the input layer;feature map with the 1/2 resolution in vertical direction while the same resolution in horizontal direction as the input layer.

Feature maps in these 3 routes are processed by normal convolution and pooling in parallel. After that, the feature map with 1/2 resolution in horizontal direction is processed by a de-convolution kernel with stride equaling to 0.5 in horizontal direction, while the feature map with 1/2 resolution in vertical direction is processed by a de-convolution kernel with stride equaling to 0.5 in vertical direction. Finally, the feature maps in 3 routes are concatenated together as the input for the following network layer. Since the feature maps are calculated with the same resolution and half resolution in one certain direction, more features of the small-scale targets in the image can be preserved. Moreover, the convolution and de-convolution with different strides are used to replace the down-sampling and up-sampling based on simple interpolation, leading to more detailed features preservation.

#### 2.3.2. Multi-Scale Pooling Strategy

Between every multi-resolution convolution block, we propose a multi-scale pooling block to replace the conventional max-pooling layer. The structure of the proposed pooling block is shown in Figure 12. The input layer is sent to 4 parallel scaling-convolution layers of which the kernel sizes are 2×2, 3×3, 4×4 and 5×5 with strides 2. These parallel convolution layers work similarly as the inception structure [18] to extract features with different receptive fields, so both global, large-scale and local, small-scale features can be obtained simultaneously. To maintain the dimensionality of feature maps and reduce the computational complexity, a 1×1 convolution layer is added to every scaling-convolution layer. Let the number of each output feature map channels from the scaling-convolution layer be *m*, the 1×1 convolution layer reduces the number of channels to m/4. Therefore, the dimensionality of the final output after concatenating these convolution layers is still *m*, leading to the comprehensive feature representation by relatively low dimensionality of feature map.

## 3. Loss Function and Training Strategy

The three sub-networks are cascaded as one end-to-end network. We train all the parameters of the uniform network once for each group of training samples, avoiding the confliction of optimizing three sub-networks separately.

### 3.1. Joint Loss of the Network

The pixel-wise loss, perceptual loss and dice loss are used as the objective of the enhancement network for parameters optimization. The joint loss function is mathematically defined as follows:(3)$$L_{j}=\mu_{1}L_{pix}+\mu_{2}L_{perc}+\mu_{3}L_{d},$$
where $L_{j}$ is the joint loss, $L_{pix}$, $L_{perc}$ and $L_{d}$ represent the pixel-wise loss, perceptual loss and dice loss respectively, $\mu_{1}$, $\mu_{2}$ and $\mu_{3}$ are the loss normalization coefficients.

The pixel-wise loss follows the traditional mean square error (MSE) loss, which can be expressed as follows:(4)$$L_{pix}=\frac{1}{WH}{∥F(y)-x∥}^{2},$$
where *y* is the input chest CT image, F(y) is the nodule detection result, *x* is the ground truth nodule map, while *W* and *H* are the width and height of the image, respectively.

The MSE loss attempts to restrain the network so that the detected nodule map could be as close to the ground truth nodule map as possible in pixel-level. However, since the nodules usually occupy small-size regions of the chest CT image, it is possible that the nodules are wrongly detected but the MSE loss is still low. To solve this problem, we need to make use of the visual perception of the CT image with nodules in it, which could make sure that the nodules are detected at the “right” regions. For the perceptual loss part, the VGG-19 network [22] is so far regarded as one of the best methods to reflect how human beings observe a given image. Specifically, in our work, the VGG-19 is used to process the image with detected pulmonary nodules and the ground truth image with true pulmonary nodules. The VGG-19 network in our algorithm can simulate how human beings observe and extract the perceptual features of the pulmonary nodules or other tissues in the chest CT image. Mathematically, the perceptual loss is defined as the Euclidean distance between the 12-th layer outputs of the VGG-19 network from generated image and the ground truth image:(5)$$L_{perc}=\frac{1}{W_{i}H_{i}}{∥\phi_{i}\left(F\left(y\right)\right)-\phi_{i}\left(x\right)∥}^{2},$$
where $W_{i}$ and $H_{i}$ denote the width and height of the convolution output of layer *i* of the VGG-19 network, and $\phi_{i}$ represents the *i*-th layer used for feature extraction. We set *i* as 12 empirically.

For the dice loss part, traditional dice loss [23] is defined to measure the similarity between the nodule-detected image F(y) and the ground truth image *x* as follows:(6)$$L_{d}=\frac{2\left|F(y)\cap x\right|}{\left|F(y)\right|+\left|x\right|},$$
where $\left|\cdot \right|$ calculates the number of non-zero pixels in the image.

However, the traditional dice loss does not work when the ground truth is negative sample, i.e., there is no nodules in the image, because the G will be null so the dice will be 0 constantly. To solve this problem, for positive samples, we use the same definition of dice loss as the traditional one, while for negative samples, we define the dice loss as the L-1 norm of the network output. Then the modified dice loss function is expressed as follows:(7)$$L_{d}=\left\{\begin{array}{cc} 1-\frac{2\left|F(y)\cap x\right|}{\left|F(y)\right|+\left|x\right|} & \;\mathrm{if}\;\left|x\right|>0\\ \left|F(y)\right| & \;\mathrm{if}\;\left|x\right|=0 \end{array}\right.,$$
where all the symbols are the same as those in Equation (Equation 6).

The process of calculating the joint loss is shown in Figure 13. The MSE loss, perceptual loss and dice loss are taken into consideration to optimize the parameters of the whole network. By minimizing the joint loss, the network can learn the differences with respect to pixel distribution, boundary shape and human perception simultaneously. Furthermore, the modified dice loss makes it possible to use negative samples for training so that the size of training database can be enlarged, leading to better pulmonary nodule detection.

### 3.2. Training Strategy

The pair-wise 2D slices of the detected nodule map F(y) from the cascaded network and the ground truth nodule map *x* are fed to the pre-trained VGG-19 network for extracting features and calculating the perceptual loss $L_{perc}$. Together with the pixel-wise loss $L_{pix}$ and the dice loss $L_{d}$, the objective loss $L_{j}$ is computed according to Equation (Equation 3). Instead of optimizing every single sub-network while fixing the parameters of the other two sub-networks, which may result in contradictory adjustments and difficulty in convergence, the loss is back-propagated to update the weights of all the parameters in the three sub-networks as the parameters of one end-to-end network, increasing the efficiency of the training process.

## 4. Experiments and Discussion

### 4.1. Experimental Dataset

To evaluate the performance of our proposed method, we use the public LUNA16 dataset [16] and the ALIBABA Cloud TianChi Medical Competition as the training and testing samples.

The LUNA16 dataset contains 888 chest CT images and 1186 pulmonary nodules. Every image represents the slice with size of 512×512 and thickness less than 2.5 mm. Pulmonary nodules in each image are annotated by four experienced radiologists during a two-phase procedure. Each radiologist annotates lesions they observed as non-nodule, nodules smaller than 3 mm, nodules larger than 3 mm. The standard of the LUNA16 challenge includes all nodules larger than 3 mm accepted by at least 3 from 4 radiologists.

The TianChi dataset contains 800 cases and the nodules are labeled by the radiologists and the form of labeling information is the same as LUNA16. The maximum slice thickness of all scans is limited to 2 mm. The nodule size distribution is as follows: 50% of them varied from 5mm to 10 mm and others were in 10 to 30 mm. Details can be seen in https://tianchi.aliyun.com/competition/entrance/231601.

### 4.2. Parameter Setting

Some key parameters for the convolution and pooling layers of the 3 detection sub-networks in series are shown in Table 2, Table 3 and Table 4, respectively. For the hyper-parameters, regularization coefficient is set to $10^{4}$, the initial learning rate is set to 0.001, $\mu_{1}$, $\mu_{2}$, $\mu_{3}$ are set to 0.9, 0.9 and 0.999 respectively. The batch size of training is set to 40, and total epoch is 100.

All the training and testing were carried out under the Pytorch framework on an Intel Core i7-4790K 4.0 GHz PC with 16G RAM and an NVIDIA TITAN XP GPU with 12G RAM.

### 4.3. Experiment Implementation

We compare our method with the three-dimensional fully convolutional neural network (3D-FCN) [12], the multi-resolution CNN (MRCNN) [13], the three-dimensional U-Net (3D-UNET) [14], the progressive resolution network with the hierarchical saliency network (PRN-HSN) [15], the DCNN pulmonary nodule detection network [24], the nodule detection with contrast limited adaptive histogram equalization (CLAHE-SVM) [25] and the Mask R-CNN-based pulmonary nodule detection network (Mask-RCNN) [26]. The testing chest CT images are processed by the proposed method, as well as the comparator state-of-the-art methods, to provide pulmonary nodule detection results. Then the detection accuracy [27], sensitivity [27], specificity [27] and the area under precision-recall (PR) curve (AUC) are calculated as follows to evaluate all the methods quantitatively:(8)$$\mathrm{Acc}=\frac{\mathrm{TP}+\mathrm{TN}}{\mathrm{TP}+\mathrm{FP}+\mathrm{TN}+\mathrm{FN}},$$
(9)$$\mathrm{Sens}=\frac{\mathrm{TP}}{\mathrm{TP}+\mathrm{FN}},$$
(10)$$\mathrm{Spec}=\frac{\mathrm{TN}}{\mathrm{TN}+\mathrm{FP}},$$
where TP denotes the number of true positive samples, i.e., the nodule samples recognized as nodules. TN represents those non-nodule samples recognized as non-nodules, FP and FN represent the non-nodule recognized as nodules and the nodule samples recognized as non-nodule samples, respectively. The precision-recall curve is generated by calculating the value of precision for every given recall.

### 4.4. Experimental Results and Discussion

#### 4.4.1. Examinations of Design Options

Figure 14(b1–b5) show several example results of the parenchyma segmentation sub-network. Table 5 shows the performance of the proposed parenchyma region segmentation sub-network, including precision, sensitivity, specificity and dice value. The average dice value of the proposed method is 0.8636, and the precision, sensitivity and specificity are 0.8792, 0.8878, and 0.9590, respectively. It can be considered that the proposed method achieves convincing performance on two datasets. The high dice value and sensitivity further prove that the segmented parenchyma mask is very close to the ground truth, which can guarantee the performance of the following nodule candidate detection.

Figure 14(c1–c5) show the results of applying the nodule candidate detection sub-network to the same input images in Figure 14(a1–a5). In addition, Figure 14(d1–d5) show the results of determining “true nodules” from the nodule candidates in Figure 14(c1–c5), respectively. Table 6 shows the performance of the proposed nodule candidate detection sub-network with respect to sensitivity and specificity. The proposed method achieves a quite high rate of sensitivity with an acceptable specificity, which demonstrates that the sub-network can detect most of the true nodules accurately with only a little over-estimation of other tiny tissues that can be further refined by the following nodule determination sub-network.

Figure 15 shows the results of detecting pulmonary nodules of using different network modules and different loss functions, including: (1) MSE loss only, (2) MSE-perceptual loss, and (3) MSE-perceptual-dice loss. Using only MSE as loss function results in failure detection of nodules with small sizes and irregular shapes. Without dice loss, the MSE-perceptual loss cannot differentiate the nodule candidates with quite small sizes because the number of pixels belonging to nodule candidates are much smaller than those belonging to background region. In contrast, combining the MSE, perceptual and dice loss can find both large and small nodule candidates, and achieve a good balance in taking the sizes and shapes of the candidates into the account of identification. Furthermore, training the three sub-networks as a uniform framework can enhance the inherent relation and co-operation of these three sub-networks in nodule detection and classification. The accuracy, sensitivity and specificity results shown in Table 7 confirm the above observation.

To further evaluate the effect of dense block in our proposed cascaded networks, we integrate the dense block in different sub-networks. Table 8 shows the results of detecting pulmonary nodules and the running time of using the dense block in: (1) the lung parenchyma segmentation sub-network solely, (2) both the lung parenchyma segmentation and the nodule candidates detection sub-networks, (3) both the lung parenchyma segmentation and the pulmonary nodule determination sub-networks, (4) all the three sub-networks. It can be considered that using the dense block in the nodule candidate detection sub-network and the pulmonary nodule determination sub-network cannot improve the detection accuracy, sensitivity and specificity to an obvious extent, but increase the average running time from 1.3411 s to 3.2354 s, 3.1563 s and 8.1823 s, respectively. It is mainly because in the lung parenchyma segmentation sub-network, there is many useful image features need to be preserved by the dense connection to distinguish the lung regions from the complex chest environment. Although in the other two cascaded sub-networks, the inputs are the parenchyma mask and the nodules regions that are relatively simple with fewer details to be preserved, using dense connection would only increase the computational burden with no obvious effect in improving the detection performance.

Figure 16 shows the changes of nodule detection accuracy and loss value over the training process of the proposed networks. Please note that the loss value is normalized in the range of 0 to 1. It demonstrates that the proposed method achieves a convergence rate after about 400 iterations.

#### 4.4.2. Comparisons with Other Methods

Figure 17 and Figure 18 illustrates the performance of pulmonary nodule detection by different methods on two example images in the LUNA16 dataset. In addition, Figure 19 and Figure 20 illustrates the performance of pulmonary nodule detection by different methods on two examples images in the TianChi dataset. As marked by green circles in Figure 18b,c and Figure 19c, the 3D-FCN and MR-CNN directly detected the nodule candidates from the original CT image without pre-processing, resulting in the incorrect determination of non-nodule tissue outside lung as nodule since the outside-lung organs are not filtered out from the nodule candidates. The 3D-UNET and PRN-HSN add the lung parenchyma region segmentation stage before detecting the nodule candidates inside-lung, so they provide better performance than 3D-FCN and MR-CNN in decreasing the over-estimation rate. However, they still suffer from unsatisfactory results for the following reasons: (1) the lung parenchyma segmentation is generated by simple thresholding with morphological operations so the near-edge regions are lost, shown as the one marked by yellow circle in Figure 20d,e; (2) the convolution kernel used in nodule candidate detection of 3D-UNET is with a small receptive field to learn global features from the image, so it is likely to confuse some small tissues as true nodules with small sizes, shown as the one marked by green circle in Figure 18d and Figure 20d; and (3) the proposed hierarchical saliency network (HSN) in PRN-HSN for nodule candidate classification omits the information with different resolutions, resulting in that the small-size nodule within the weakened, low-resolution region cannot be correctly recognized, as shown by the yellow circle in Figure 18e. The DCNN method simply applies the Faster RCNN method to provide good performance with low computational cost, but it may omit the nodules on the parenchymal edge shown as yellow circles in Figure 20f. CLAHE-SVM method adds a contrast-enhancement pre-processing before the nodule detection, leads to better performance on detecting nodules in the low-contrast region. However, it is easily to over-enhance the small-size tissues and over-estimate them as nodules, as shown by the green circles in Figure 19g andFigure 20g. The detection is also implemented over the whole image, so the nodule on the parenchyma edge may be under-estimated show by the yellow circle in Figure 20g. The Mask-RCNN method provides better effects than the above methods because of the good performance of Mask-RCNN in object detection. However, the performance is not stable for the small-size tissues and the irregular-shape nodule, shown by the green circles in Figure 17h and Figure 19h, and the yellow circle in Figure 19h. The proposed method takes the advantage of a series of U-Net-like networks to perform the nodule detection following a “coarse-to-fine” order of inside-lung region detection, nodule candidate detection and nodule determination. The U-Net network is modified by embedding inception structure, replacing the convolution and pooling by dilated convolution, and adapting multi-scale pooling and multi-resolution convolution connection, for different requirements of the three stages, respectively. Moreover, it makes use of the MSE loss, VGG-19-based perceptual loss as the complement of dice loss to optimize the whole framework. Therefore, as shown in Figure 17i, Figure 18i, Figure 19i and Figure 20i, the proposed framework provides superior performance on pulmonary nodule detection with low over-estimation of non-nodule tissues at the same time.

Figure 21 and Figure 22 show the Precision-Recall curves of applying different methods to detect the nodules in the testing dataset of LUNA16 dataset and TianChi dataset, while Table 9 and Table 10 illustrate the values of AUC and average accuracies, sensitivities and specificities of these methods. It can be considered that the proposed method provides highest accuracy, sensitivity, specificity and AUC, which confirms our qualitative observations.

#### 4.4.3. Running Time

The running time of implementing different methods on the testing data is shown in Table 11. It costs our proposed method 1.3411 s on average to generate the nodule detection results for a CT scan image. The running time is a little longer than MR-CNN, PRN-HSN, DCNN and CLAHE-SVM, but it is acceptable considering that our method provides 2.49%, 1.12%, 2.01% and 1.30% higher detection accuracy than them.

#### 4.4.4. Failure Case

The proposed method performs effectively on most cases of pulmonary nodule detection. However, when the nodule is on the edge of lung parenchyma and the intensity level of nodule is quite close to that of the parenchyma, our method cannot distinguish the nodule from the outside-lung region. When the inside-lung tissue is with the medium size and similar shape as the nodule, our method is likely to over-estimate it as a nodule. Both cases are shown in Figure 23.

## 5. Conclusions

In this paper, we novelly proposed to detect pulmonary nodules from chest CT images through a uniform framework consisting of three consecutive U-Net-like networks. An inception structure is used to replace the first convolution layer of the U-Net to estimate the lung parenchyma region. Then another U-Net-like network is proposed by leveraging the dilated convolution to replace all the convolution layers to detect the small tissues as nodule candidates. Finally, the third U-Net-like network is proposed by adapting multi-scale pooling and multi-resolution convolution connection to determine the true nodules. Moreover, the three sub-networks are integrated together and optimized using a fused loss function consisting of MSE loss, perceptual loss and dice loss. Experimental results demonstrate that the proposed method provides superior performance of pulmonary nodule detection to the state-of-the-art methods on the public LUNA16 dataset and the TianChi competition dataset.

## Figures and Tables

**Figure 1 sensors-20-04301-f001:**
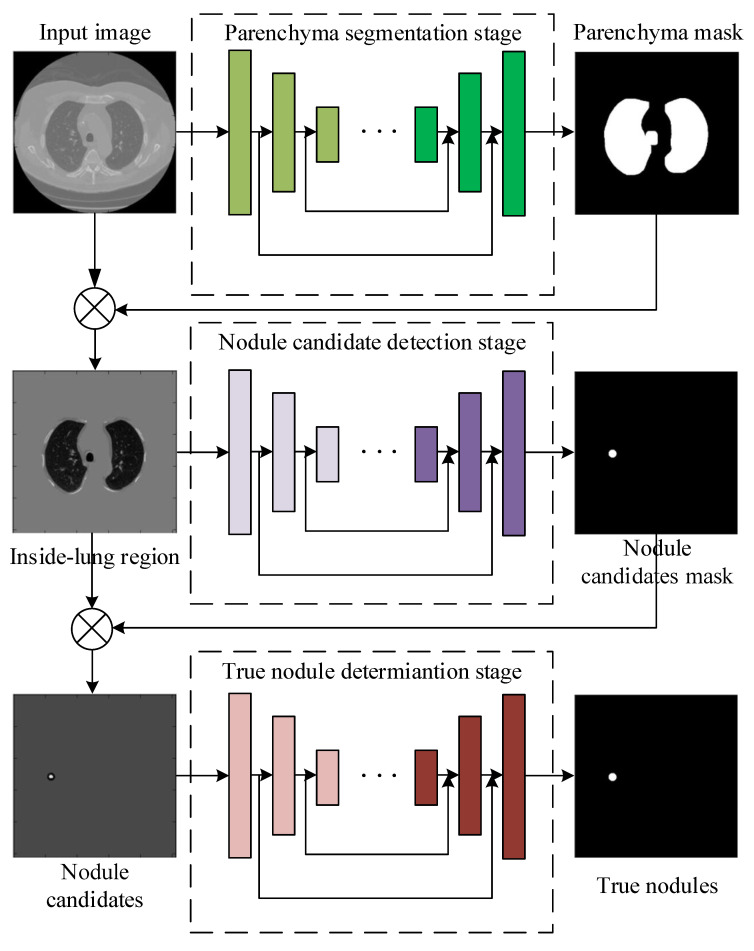
The framework of the proposed pulmonary nodule detection method by cascading three sub-networks for lung parenchyma segmentation, nodule candidate detection and true nodule determination.

**Figure 2 sensors-20-04301-f002:**
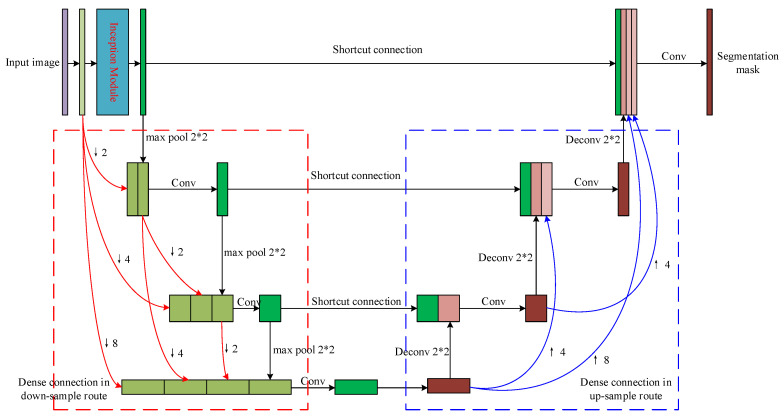
The structure of the proposed sub-network for lung parenchyma segmentation.

**Figure 3 sensors-20-04301-f003:**
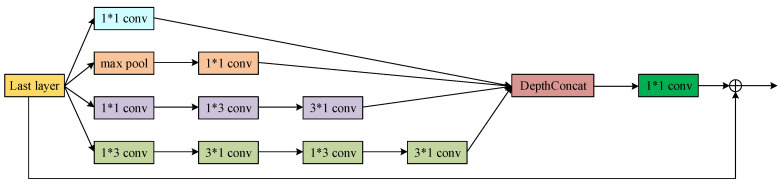
The modified inception module used in the proposed lung parenchyma segmentation sub-network.

**Figure 4 sensors-20-04301-f004:**
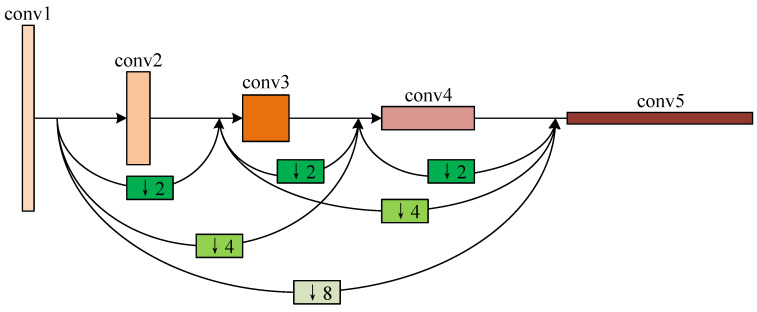
The dense connection of the 4 convolutional layers in the proposed lung parenchyma segmentation sub-network.

**Figure 5 sensors-20-04301-f005:**
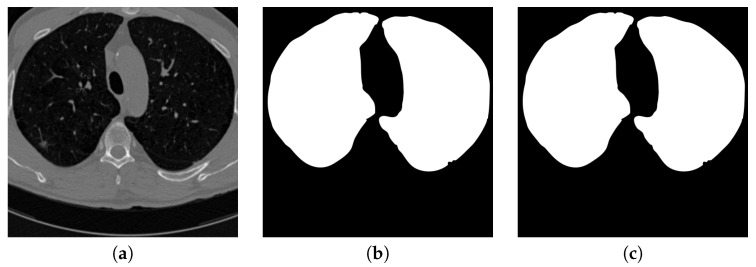
An example of segmenting parenchyma mask from the chest CT image. (**a**) is the input chest CT image, (**b**) is the lung parenchyma mask detected by the proposed parenchyma detection sub-network, (**c**) is the ground truth lung parenchyma mask.

**Figure 6 sensors-20-04301-f006:**
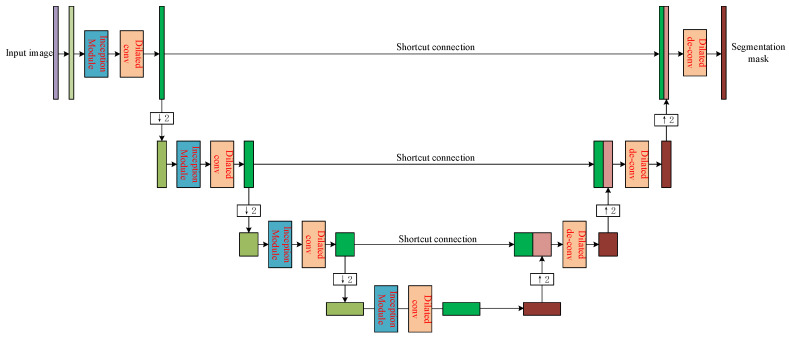
The structure of the proposed sub-network for nodule candidate region detection.

**Figure 7 sensors-20-04301-f007:**
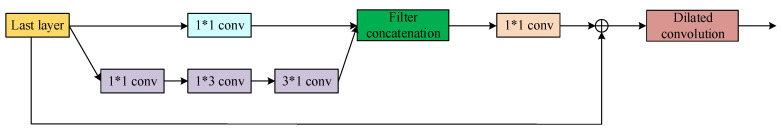
The general structure of the convolution block used in the nodule candidate detection sub-network.

**Figure 8 sensors-20-04301-f008:**
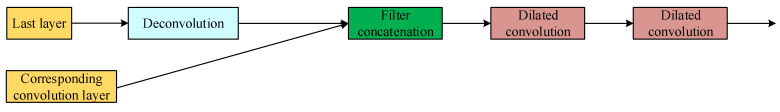
The general structure of the de-convolution block used in the nodule candidate detection sub-network.

**Figure 9 sensors-20-04301-f009:**
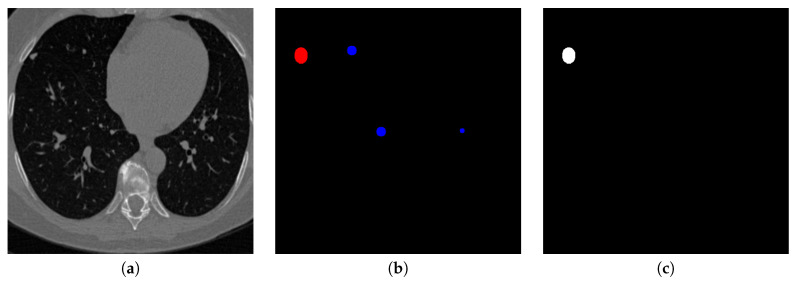
An example of detecting nodule candidates from the chest CT image. (**a**) is the input chest CT image, (**b**) is the nodule candidates detected by the proposed nodule candidate detection sub-network, (**c**) is the ground truth nodule region.

**Figure 10 sensors-20-04301-f010:**
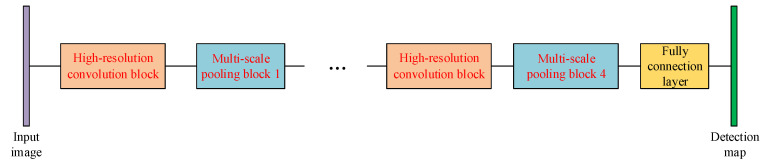
The structure of the proposed sub-network for nodule determination.

**Figure 11 sensors-20-04301-f011:**
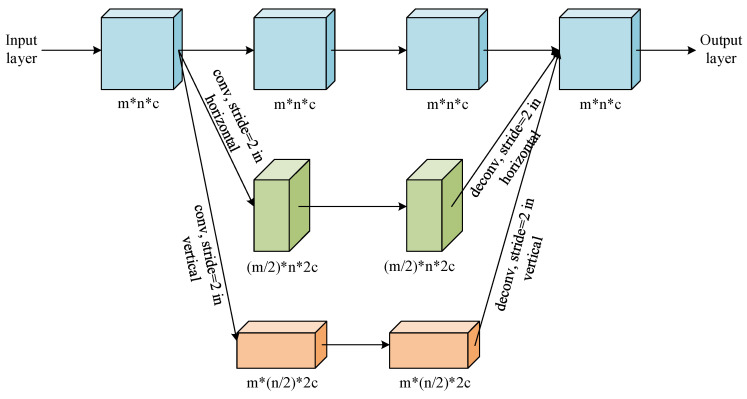
The multi-resolution feature concatenation module of the proposed determination sub-network.

**Figure 12 sensors-20-04301-f012:**
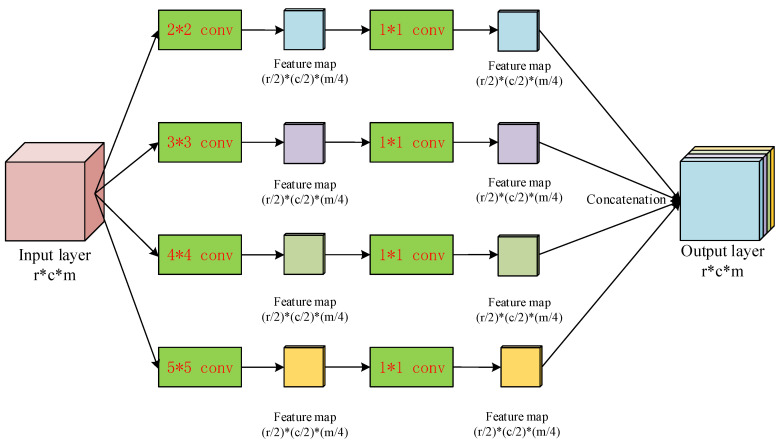
The multi-scale pooling block used in the proposed determination sub-network.

**Figure 13 sensors-20-04301-f013:**
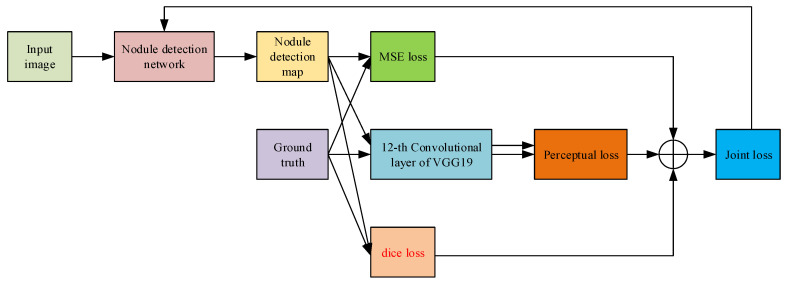
The flow of calculating the joint loss for optimizing the parameters of the nodule detection network.

**Figure 14 sensors-20-04301-f014:**
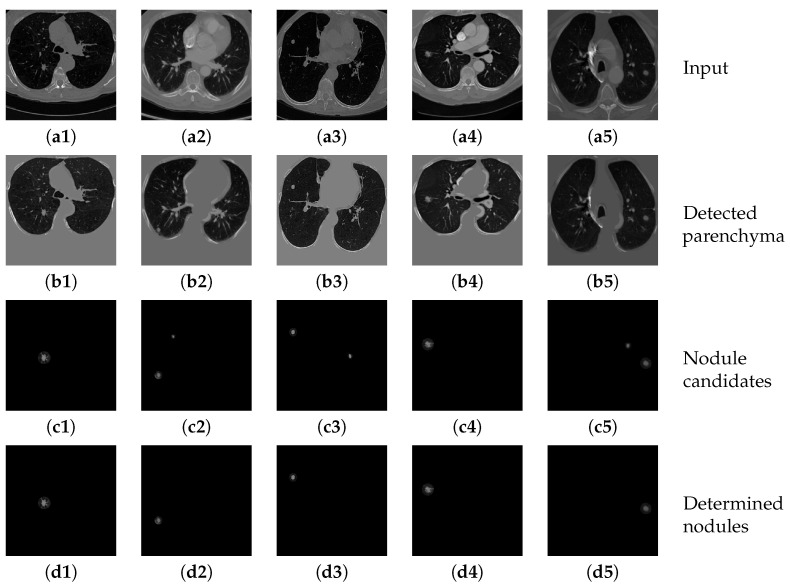
The results of detecting parenchyma regions, nodule candidates and determining nodules by the proposed cascaded network on five different chest CT images.

**Figure 15 sensors-20-04301-f015:**
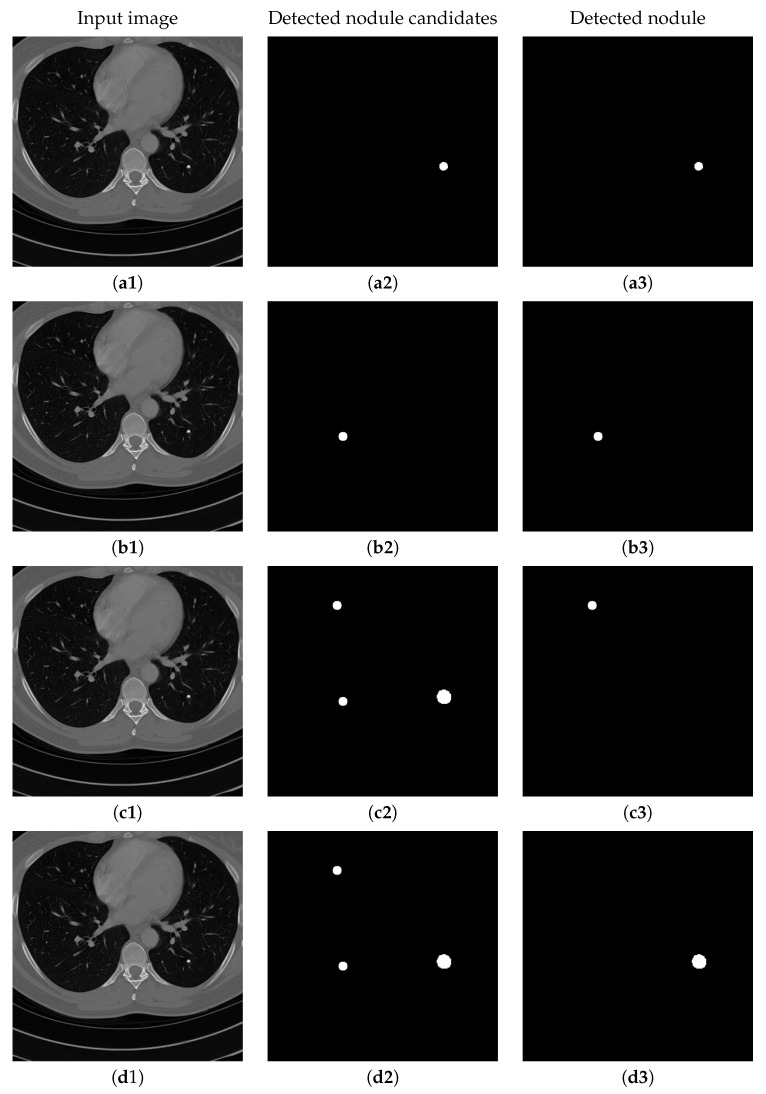
The results of detecting pulmonary nodule through the proposed network with different loss strategies. Columns from left to right are: input images, detected nodule candidates, detected nodules. Rows from top to bottom are: ground truth detection results, nodule candidates and nodule detection by the proposed network with MSE loss, MSE-perceptual loss and the MSE-perceptual-dice loss.

**Figure 16 sensors-20-04301-f016:**
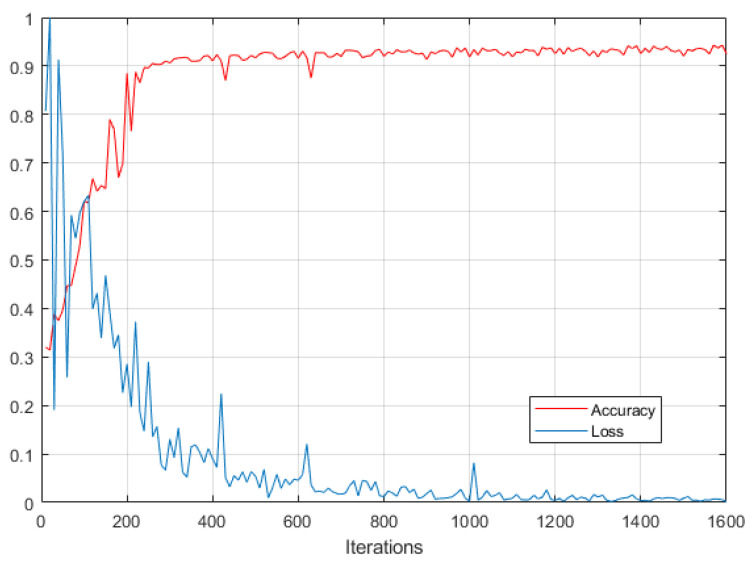
Nodule detection accuracy and loss over the training process of the proposed networks.

**Figure 17 sensors-20-04301-f017:**
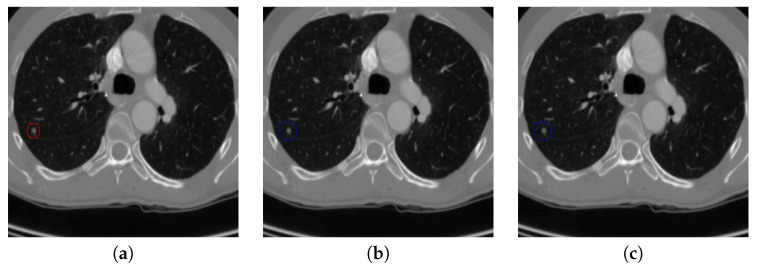
The result of detecting pulmonary nodules by different methods on one example image in LUNA16 dataset. Red circle represents the ground truth region of nodules, blue circles represent the correct estimation of pulmonary nodules, green circles represent the over-detected nodules, while yellow circles denote the nodules being omitted. (**a**–**i**) are: ground truth nodule in the given chest CT image, nodule detected by 3D-FCN, MR-CNN, 3D-UNET, PRN-HSN, DCNN, CLAHE-SVM, MASK-RCNN and our proposed method.

**Figure 18 sensors-20-04301-f018:**
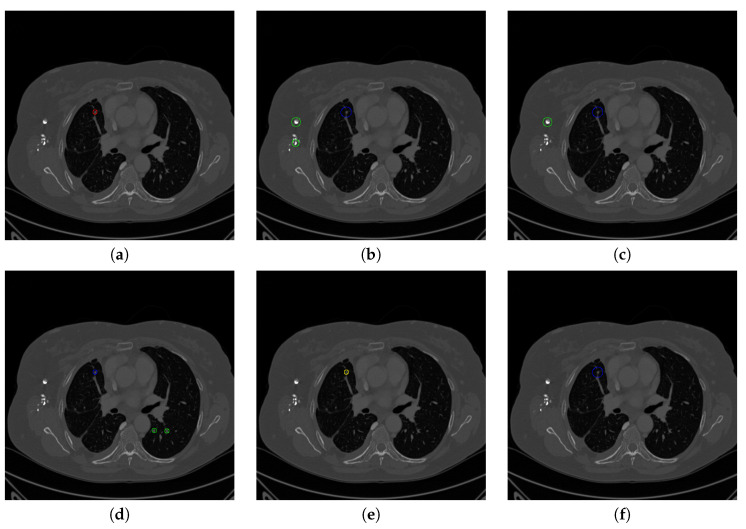
The result of detecting pulmonary nodules by different methods on another example image in LUNA16 dataset. Red circle represents the ground truth region of nodules, blue circles represent the correct estimation of pulmonary nodules, green circles represent the over-detected nodules, while yellow circles denote the nodules being omitted. (**a**–**i**) are: ground truth nodule in the given chest CT image, nodule detected by 3D-FCN, MR-CNN, 3D-UNET, PRN-HSN, DCNN, CLAHE-SVM, MASK-RCNN and our proposed method.

**Figure 19 sensors-20-04301-f019:**
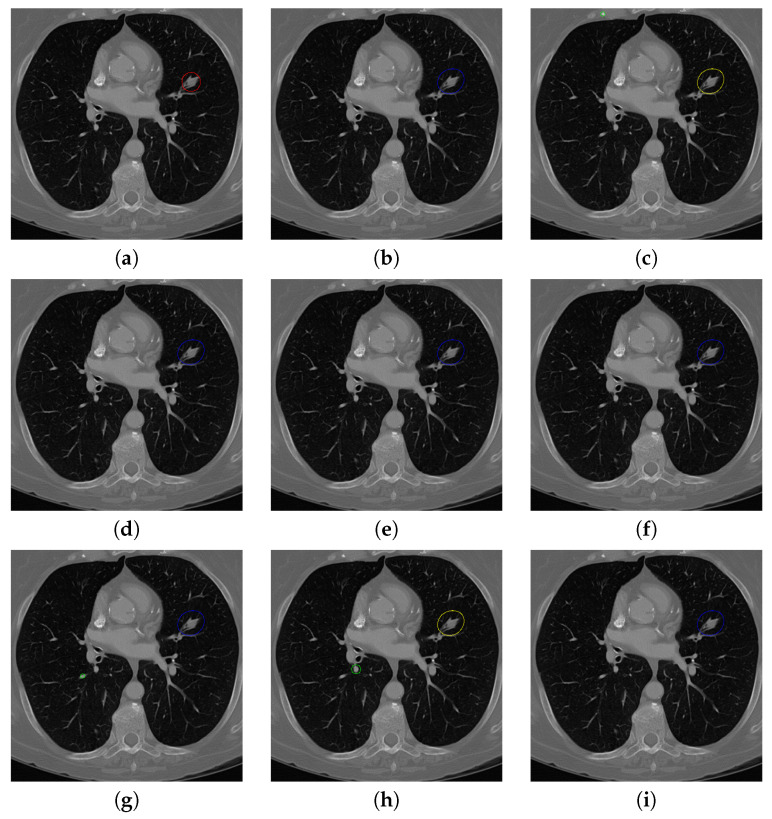
The result of detecting pulmonary nodules by different methods on one example image in TianChi dataset. Red circle represents the ground truth region of nodules, blue circles represent the correct estimation of pulmonary nodules, green circles represent the over-detected nodules, while yellow circles denote the nodules being omitted. (**a**–**i**) are: ground truth nodule in the given chest CT image, nodule detected by 3D-FCN, MR-CNN, 3D-UNET, PRN-HSN, DCNN, CLAHE-SVM, MASK-RCNN and our proposed method.

**Figure 20 sensors-20-04301-f020:**
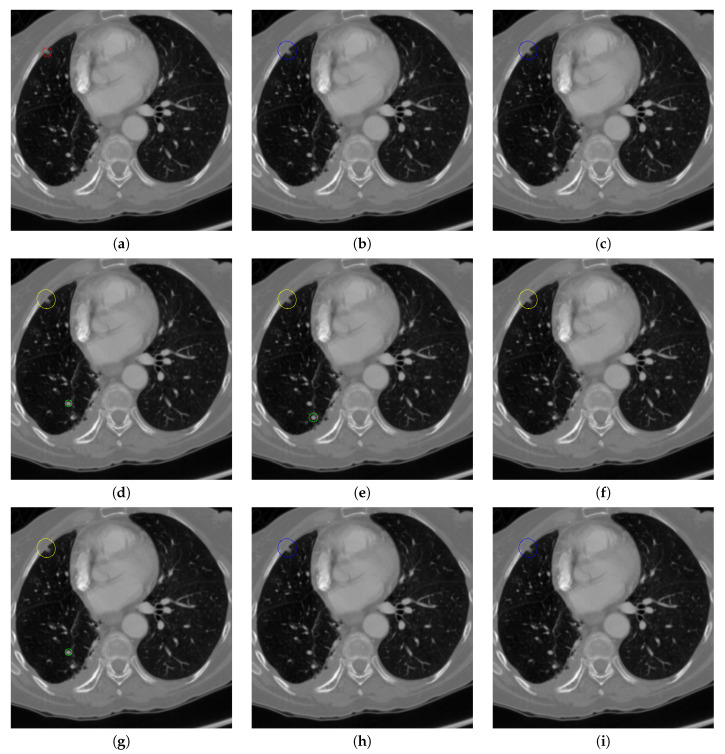
The result of detecting pulmonary nodules by different methods on another example image in TianChi dataset. Red circle represents the ground truth region of nodules, blue circles represent the correct estimation of pulmonary nodules, green circles represent the over-detected nodules, while yellow circles denote the nodules being omitted. (**a**–**i**) are: ground truth nodule in the given chest CT image, nodule detected by 3D-FCN, MR-CNN, 3D-UNET, PRN-HSN, DCNN, CLAHE-SVM, MASK-RCNN and our proposed method.

**Figure 21 sensors-20-04301-f021:**
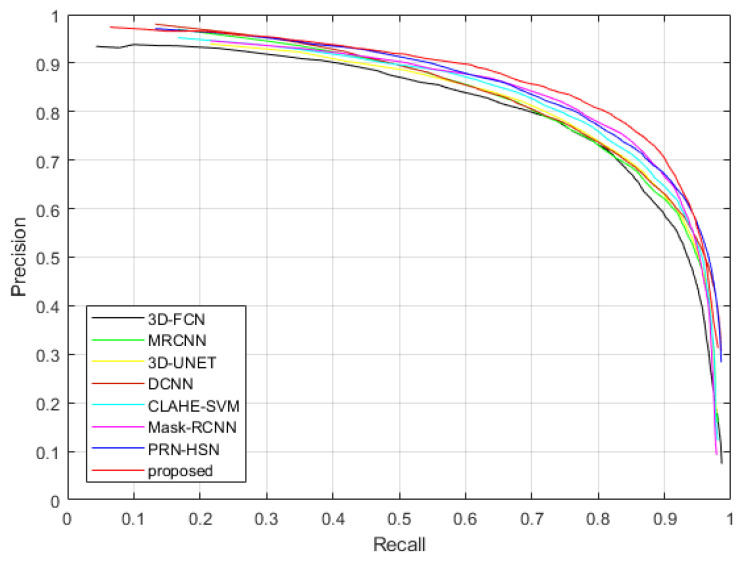
The Precision-Recall curves of applying different methods to detect nodules in the testing dataset of LUNA16 dataset.

**Figure 22 sensors-20-04301-f022:**
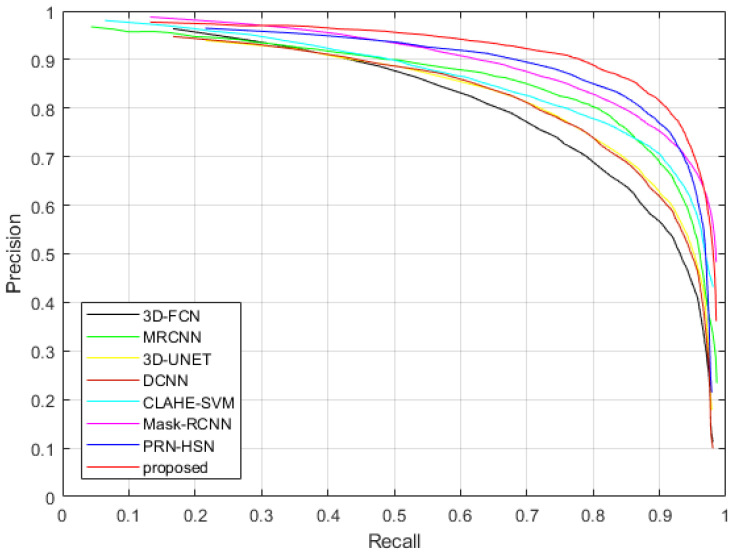
The Precision-Recall curves of applying different methods to detect nodules in the testing dataset of TianChi dataset.

**Figure 23 sensors-20-04301-f023:**
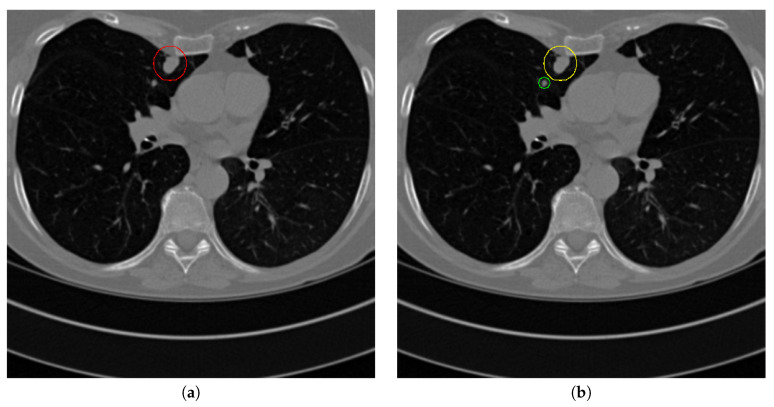
Failure detection case. (**a**) is the ground truth nodule detection result, (**b**) is the nodule detection by the proposed method. Red circle represents the ground truth region of nodule, green circle represents the over-detected nodule, while yellow circle denotes the nodule being omitted.

**Table 1 sensors-20-04301-t001:** Advantages and disadvantages of the comparison methods.

Methods	Advantages	Disadvantages
3D-FCN	Simple implementation	Incorrect determination of non-nodule tissue outside lung as nodule
MRCNN	Simple implementation, multi-resolution model is good for small-size nodule detection	Incorrect determination of non-nodule tissue outside lung as nodule
3D-UNET	Lung parenchyma regions are first detected, lower over-estimation rate	Near-edge regions are easily lost, confusion of small tissues as nodules
PRN-HSN	Lung parenchyma regions are first detected, lower over-estimation rate	Near-edge regions are easily lost, small-size nodule within weakened, low-resolution region cannot be recognized
DCNN	Simple implementation	Small-size nodules on the edges of parenchyma region are easily omitted
CLAHE-SVM	Nodules within low-contrast regions can be detected with the contrast-enhancement pre-processing	Small-size tissues are easily over-estimated as nodules
Mask-RCNN	Small-size nodules can be detected accurately, nodule detection and segmentation are achieved simultaneously	Heavy computational cost, unstable performance on nodule detection
proposed	Pulmonary nodules can be detected accurately with low over-estimation of non-nodule tissues	Nodules within low-contrast, vague regions cannot be detected

**Table 2 sensors-20-04301-t002:** Some key parameters in the lung parenchyma region segmentation sub-network

Modules		Parameters
U-Net	Conv1	3×3×128, stride=2
	Conv2	3×3×256, stride=2
	Conv3	3×3×512, stride=2
	Deconv1	3×3×512, stride=2
	Deconv2	3×3×256, stride=2
	Deconv3	3×3×128, stride=2
	Deconv4	3×3×64, stride=2
Inception	Conv1	1×1×16, stride=1
	Conv2	1×1×16, stride=1
	Conv3	1×1×16, stride=1
	Conv4	1×3×16, stride=1
	Conv5	3×1×16, stride=1
	Conv6	1×3×16, stride=1
	Conv7	3×1×16, stride=1
	Conv8	1×3×16, stride=1
	Conv8	3×1×16, stride=1
	Conv8	1×1×64, stride=1

**Table 3 sensors-20-04301-t003:** Some key parameters in the nodule candidate detection sub-network (*m* represents the channels of the former layer)

Modules		Parameters
U-Net	Conv1	3×3×64, stride=1
	Conv2	3×3×128, stride=2
	Conv3	3×3×256, stride=2
	Conv4	3×3×512, stride=2
	Deconv1	3×3×512, stride=2
	Deconv2	3×3×256, stride=2
	Deconv3	3×3×128, stride=2
	Deconv4	3×3×64, stride=2
Inception	Conv1	1×1×(m/2), stride=1
	Conv2	1×1×(m/2), stride=1
	Conv3	1×3×(m/2), stride=1
	Conv4	3×1×(m/2), stride=1
	Conv5	1×1×m, stride=1
Dilated convolution	Di-conv1	3×3×m, stride=1, dilationrate=2

**Table 4 sensors-20-04301-t004:** Some key parameters in the nodule determination sub-network (*m* and *c* represent the channels of the corresponding former layers)

Modules		Parameters
Multi-resolution convolution	Conv1	3×3×c, stride=(1,1)
	Conv2	3×3×(2c), stride=(2,1)
	Conv3	3×3×(2c), stride=(1,2)
	Deconv1	3×3×c, stride=(1,1)
	Deconv2	3×3×c, stride=(2,1)
	Deconv3	3×3×c, stride=(1,2)
Multi-scale pooling	Conv1	2×2×(m/4), stride=2
	Conv2	3×3×(m/4), stride=2
	Conv3	4×4×(m/4), stride=2
	Conv4	5×5×(m/4), stride=1
	Conv5	1×1×(m/4), stride=1
	Conv6	1×1×(m/4), stride=1
	Conv7	1×1×(m/4), stride=1
	Conv8	1×1×(m/4), stride=1

**Table 5 sensors-20-04301-t005:** Quantitative results of detecting parenchyma regions from input chest CT images by the proposed network.

Method	Precision	Sensitivity	Specificity	Dice
Inception-dense U-Net sub-network	0.8792	0.8878	0.9590	0.8636

**Table 6 sensors-20-04301-t006:** Quantitative results of detecting nodule candidates from input chest CT images by the proposed network.

Method	Sensitivity	Specificity
Dilated-convolution U-Net sub-network	0.9692	0.9078

**Table 7 sensors-20-04301-t007:** Quantitative results associated with different combinations of losses of the proposed network for detecting nodules from the image

Methods	Accuracy	Sensitivity	Specificity
MSE	0.9182	0.8502	0.9315
MSE-perceptual	0.9237	0.8748	0.9327
MSE-perceptual-dice	0.9390	0.8988	0.9476

**Table 8 sensors-20-04301-t008:** Results of detecting pulmonary nodules and running time of using dense block in different sub-networks.

Methods	Accuracy	Sensitivity	Specificity	Running Time (s)
Dense block in parenchyma segmentation sub-network	0.9390	0.8988	0.9476	1.3411
Dense block in parenchyma segmentation and nodule candidate detection sub-networks	0.9392	0.8990	0.9480	3.2354
Dense block in parenchyma segmentation and nodule determination sub-networks	0.9395	0.8985	0.9486	3.1563
Dense block in all three sub-networks	0.9396	0.8992	0.9485	8.1823

**Table 9 sensors-20-04301-t009:** Quantitative results associated with different nodule detection methods on LUNA16 dataset.

Methods	Accuracy	Sensitivity	Specificity	AUC
3D-FCN	0.8975	0.8261	0.9295	0.9027
MRCNN	0.9162	0.8487	0.9371	0.9173
3D-UNET	0.9176	0.8516	0.9394	0.9259
PRN-HSN	0.9286	0.8782	0.9430	0.9371
DCNN	0.9170	0.8496	0.9388	0.9195
CLAHE-SVM	0.9251	0.8672	0.9415	0.9328
Mask-RCNN	0.9291	0.8806	0.9447	0.9396
proposed	0.9390	0.8988	0.9476	0.9615

**Table 10 sensors-20-04301-t010:** Quantitative results associated with different nodule detection methods on TianChi dataset.

Methods	Accuracy	Sensitivity	Specificity	AUC
3D-FCN	0.9087	0.8358	0.9387	0.9136
MRCNN	0.9259	0.8568	0.9468	0.9261
3D-UNET	0.9297	0.8607	0.9498	0.9339
PRN-HSN	0.9356	0.8853	0.9547	0.9462
DCNN	0.9288	0.8598	0.9485	0.9287
CLAHE-SVM	0.9353	0.8781	0.9542	0.9407
Mask-RCNN	0.9389	0.8913	0.9596	0.9579
proposed	0.9475	0.9036	0.9655	0.9722

**Table 11 sensors-20-04301-t011:** The running time of applying different methods in detecting nodules.

**Methods**	3D-FCN	MRCNN	3D-UNET	PRN-HSN	DCNN	CLAHE-SVM	Mask-RCNN	proposed
**Time(s)**	1.9134	1.0975	1.6324	1.1270	1.0753	1.0572	1.5985	1.3411

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
