# Peer review of "A Novel Pulmonary Nodule Detection Model Based on Multi-Step Cascaded Networks"

_sensors, 2020, doi:10.3390/s20154301_

Round 1
Reviewer 1 Report
This paper focuses on the problem to of lung cancer nodule detection. A cascaded framework is proposed to solve the problem in a coarse-to-fine manner. The authors also introduced some multi-scale methods to deal with the size variation among targets. Experiments on the LIDC-IDRI dataset show impressive improvement compared to the state-of-the-art peers. However, there are some shortcomings and questions listed below:
- The proposed network is composed of three U-Net-like subnetworks, each of which is assigned with specific target, but there is only supervision on the last subnetwork. Therefore, I’m wondering that how could the first two networks learn accurately without supervision. I think it necessary to evaluate the accuracy of them to prove that the network indeed works as the way you described.
- Some design of the network structure should be explained or conducted ablation studies on. For example, the dense connection is used in the lung parenchyma segmenting network but not in the other two. Now that its motivation is avoid the incompleteness, why isn’t it used in the subsequent operations?
- In your perceptual loss part another network is introduced to measure the difference of the detection result with the ground truth, which I didn’t realize much necessity. I think an easier module or branch should be considered to replace it, otherwise its necessity should be proved.
- The experimental part needs to be strengthened. The output and the effects of the proposed parts should be visualized and verified, but the shown version lacks of them thus is less convincing.
- It’s strongly suggested to remove the redundant parts for better reading experience. There is a dense repetition of descriptions in this paper, which should be removed for better reading experience. And some figures can be merged or rearranged since they carry little information.
- On account of the common problem you solve, the literatures listed below should be compared with and added to the reference:
[1] Hongtao Xie, Dongbao Yang, Nannan Sun, Zhineng Chen, Yongdong Zhang: Automated pulmonary nodule detection in CT images using deep convolutional neural networks. Pattern Recognit. 85: 109-119 (2019)
[2] Khan, S. A., Hussain, S., Yang, S., & Iqbal, K. (2019). Effective and Reliable Framework for Lung Nodules Detection from CT Scan Images. Scientific reports, 9(1), 1-14.
- Some typos and errors in the formulas should be avoided.
Author Response
Point 1: The proposed network is composed of three U-Net-like subnetworks, each of which is assigned with specific target, but there is only supervision on the last subnetwork. Therefore, I’m wondering that how could the first two networks learn accurately without supervision. I think it necessary to evaluate the accuracy of them to prove that the network indeed works as the way you described. 

Response 1: We have added, in Fig. 14, Table 4 and 5 in Sec. 4.4.1, to show and evaluate the accuracy of the lung parenchyma segmentation sub-network and the nodule candidates detection sub-network. It is shown that the first two sub-networks can provide good segmentation and detection effects. It is because that the output of the current sub-network is used as the input of the next sub-network, and the connection between the three sub-networks makes is possible to train them as one uniform network.
Point 2: Some design of the network structure should be explained or conducted ablation studies on. For example, the dense connection is used in the lung parenchyma segmenting network but not in the other two. Now that its motivation is avoid the incompleteness, why isn’t it used in the subsequent operations?
Response 2: We have conducted, in the 4-th paragraph in Sec. 4.4.1, the ablation studies on using dense-block in different sub networks. We have shown the results of detecting nodules and running time of using dense block in different sub-networks in Table 7. It can be considered that using the dense block in the nodule candidates detection sub-network and the pulmonary nodule determination sub-network cannot improve the detection accuracy, sensitivity and specificity to an obvious extent, but increase the average running time from 1.3411s to 3.2354s, 3.1563s and 8.1823s, respectively. It is mainly because in the lung parenchyma segmentation sub-network, there is many useful image features need to be preserved by the dense connection to distinguish the lung regions from the complex chest environment. While in the other two cascaded sub-network, the inputs are the parenchyma mask and the nodules regions that are relatively simple with fewer details to be preserved, so using dense connection would only increase the computational burden with no obvious effect in improving the detection performance.
Point 3: In your perceptual loss part another network is introduced to measure the difference of the detection result with the ground truth, which I didn’t realize much necessity. I think an easier module or branch should be considered to replace it, otherwise its necessity should be proved.
Response 3: We have clarified, in the 3rd paragraph in Sec. 3.1, the necessity of using VGG-19 network to extract perceptual features for calculating the perceptual loss. The VGG-19 network is so far regarded as one of the best methods to reflect how human beings observe a given image. Therefore, using VGG-19 can make use of the visual perception of the CT image with nodules in it, which could make sure that the nodules are detected at the ``right'' regions.
Point 4: The experimental part needs to be strengthened. The output and the effects of the proposed parts should be visualized and verified, but the shown version lacks of them thus is less convincing.
Response 4: We have strengthened the experimental part by showing the results of every sub-network in Fig. 14 and Fig. 15. We also added more cases of nodule detection from another chest CT image dataset in Fig. 18.
Point 5: It’s strongly suggested to remove the redundant parts for better reading experience. There is a dense repetition of descriptions in this paper, which should be removed for better reading experience. And some figures can be merged or rearranged since they carry little information.
Response 5: We have removed some redundant text and merged some figures in the manuscript for better reading experience.
Point 6: On account of the common problem you solve, the literatures listed below should be compared with and added to the reference:
[1] Hongtao Xie, Dongbao Yang, Nannan Sun, Zhineng Chen, Yongdong Zhang: Automated pulmonary nodule detection in CT images using deep convolutional neural networks. Pattern Recognit. 85: 109-119 (2019)
[2] Khan, S. A., Hussain, S., Yang, S., & Iqbal, K. (2019). Effective and Reliable Framework for Lung Nodules Detection from CT Scan Images. Scientific reports, 9(1), 1-14.
Response 6: We have added the above two methods in the experiment part and compared their performance with our proposed method.
Point 7: Some typos and errors in the formulas should be avoided.
Response 7: We have corrected typos and errors in the manuscript.

Reviewer 2 Report
In this paper, the authors proposed an “A novel pulmonary nodule detection model based on multi-step cascaded networks”. The framework proposed in the paper is comprised of three cascaded networks. The first cascaded network is based on U-Net based on the integration of inception structure and dense skip connections. The second is based on the concept of dilated convolution and the third is based on multi-scale pooling and multi-resolution convolution connection. The experimental results of the proposed framework illustrated that it is efficient than other methods in challenging issues. However, there are some comments that are listed as follows.
- Please include the comparison of your method and previous methods in the form of a table. Your comparison table should include a short description of the methods, strengths, and weaknesses of your method and previous methods.
- Please include the training accuracy and loss curves in your results.
- Results are tested on only one publicly available dataset. Please try to check the results on other datasets.
- Please include the most recent papers (2019-2020) in your comparisons. There are 2-3 papers from 2019 but not a single paper from 2020.
- Please include the image results after each cascaded stage. So that the strength of your method can be visualized. Also, include good and bad detected cases with your method and previous methods on the same images.
Author Response
Point 1: Please include the comparison of your method and previous methods in the form of a table. Your comparison table should include a short description of the methods, strengths, and weaknesses of your method and previous methods. 

Response 1: We have included the comparison of our method and previous methods in Table 8.
Point 2: Please include the training accuracy and loss curves in your results.
Response 2: We have included the training accuracy and loss curve of our proposed method in Fig. 16.
Point 3: Results are tested on only one publicly available dataset. Please try to check the results on other datasets.
Response 3: We have performed all the comparison methods on another publicly available dataset, that is, Alibaba Cloud TianChi Medical Competition. The qualitative results of the TianChi dataset are shown in Fig. 18, the quantitative results are shown in Table 10, and the precision-recall curves are shown in Fig. 20.
Point 4: Please include the most recent papers (2019-2020) in your comparisons. There are 2-3 papers from 2019 but not a single paper from 2020.
Response 4: We have included three recent papers in the comparisons. The methods are as follows:
- Hongtao Xie, Dongbao Yang, Nannan Sun, Zhineng Chen, Yongdong Zhang: Automated pulmonary nodule detection in CT images using deep convolutional neural networks. Pattern Recognition. 85: 109-119 (2019).
- Khan, S. A., Hussain, S., Yang, S., & Iqbal, K. (2019). Effective and Reliable Framework for Lung Nodules Detection from CT Scan Images. Scientific reports, 9(1), 1-14.
- Cai, L.; Long, T.; Dai, Y.; Huang, Y. Mask R-CNN-Based Detection and Segmentation for Pulmonary Nodule 3D Visualization Diagnosis. IEEE Access 2020, 8, 44400 – 44409.
The above three methods have been implemented and applied to both LUNA16 dataset and TianChi dataset. The results in Fig. 17, Fig. 18, Fig. 19, Fig. 20, Table 9 and Table 10 have been updated.
Point 5: Please include the image results after each cascaded stage. So that the strength of your method can be visualized. Also, include good and bad detected cases with your method and previous methods on the same images.
Response 5: We have included the images results after each cascaded stage in Fig. 14 and Fig. 15. We have also included bad detected case of our proposed method in Fig. 21 with some discussions of the failure reason.

Round 2
Reviewer 1 Report
The authors have well addressed my concerns. However, the paper needs English polishing.
Reviewer 2 Report
1) Comparison table 8 should be after the introduction and related work.
2) The training curve for accuracy is included the loss curve is missing. Please include both the curves in the same figure to avoid an unnecessary increase in the paper's length.
3) Please try to reduce the number of sample images in Figures 17 and 18 as the figures are very small and it is difficult to see the blue, green, and yellow circles around nodules.
4) Point 5 in the previous review was incorrectly addressed because filter output images were required after each stage.
5) Please check spellings and English grammar.
noudle > nodule (line 214)
Round 3
Reviewer 2 Report
Thank you for making required changes. Congratulations to authors.